# Circulating microRNAs in Cerebrospinal Fluid and Plasma: Sensitive Tool for Detection of Secondary CNS Involvement, Monitoring of Therapy and Prediction of CNS Relapse in Aggressive B-NHL Lymphomas

**DOI:** 10.3390/cancers14092305

**Published:** 2022-05-06

**Authors:** Pavle Krsmanovic, Heidi Mocikova, Kamila Chramostova, Magdalena Klanova, Marie Trnkova, Michal Pesta, Peter Laslo, Robert Pytlik, Tomas Stopka, Marek Trneny, Vit Pospisil

**Affiliations:** 1Institute of Pathological Physiology, 1st Faculty of Medicine, Charles University, 128 53 Prague, Czech Republic; pavle.krsmanovic@lf1.cuni.cz (P.K.); kamila.chramostova@lf1.cuni.cz (K.C.); magdalena.klanova@lf1.cuni.cz (M.K.); 2Department of Haematology, University Hospital Kralovske Vinohrady and 3rd Faculty of Medicine, Charles University, 100 34 Prague, Czech Republic; heidi.mocikova@fnkv.cz; 31st Department of Medicine, Charles University General Hospital, 128 08 Prague, Czech Republic; trnkova@lymphoma.cz (M.T.); tomas.stopka@lf1.cuni.cz (T.S.); marek.trneny@vfn.cz (M.T.); 4Faculty of Mathematics and Physics, Charles University, 186 75 Prague, Czech Republic; michal.pesta@mff.cuni.cz; 5Section of Experimental Haematology, Leeds Institute of Cancer and Pathology, St James’s University Hospital, University of Leeds, Leeds LS2 9JT, UK; p.laslo@leeds.ac.uk; 6Department of Cell Therapy, Institute of Haematology and Blood Transfusion, 128 20 Prague, Czech Republic; robert.pytlik@uhkt.cz; 7Biocev, 1st Faculty of Medicine, Charles University, 252 50 Vestec, Czech Republic

**Keywords:** CNS, lymphoma, microRNA, DLBCL, MCL, Burkitt, B-NHL, relapse, cerebrospinal fluid, plasma

## Abstract

**Simple Summary:**

Lymphoma involving the central nervous system and CNS relapse present diagnostic and predictive challenges. Its diagnosis is based on conventional methods with low sensitivity and/or specificity. More powerful tools for its early detection, response evaluation, and CNS relapse prediction are needed. MicroRNAs are short post-transcriptional gene regulators that are remarkably stable and detectable extracellularly in body fluids. We evaluated the diagnostic and predictive potential of circulating oncogenic microRNAs (oncomiRs) in CSF and plasma for the detection of secondary CNS involvement in aggressive B-NHL lymphomas, as well as for detection and prediction of their CNS relapse. Our findings indicate that the evaluation of oncogenic microRNAs in CSF and plasma potentially provides a sensitive tool for the early detection of secondary CNS lymphoma, the monitoring and estimating of treatment efficacy, and the prediction and early detection of CNS relapse.

**Abstract:**

Lymphoma with secondary central nervous system (CNS) involvement represents one of the most aggressive malignancies, with poor prognosis and high mortality. New diagnostic tools for its early detection, response evaluation, and CNS relapse prediction are needed. We analyzed circulating microRNAs in the cerebrospinal fluid (CSF) and plasma of 162 patients with aggressive B-cell non-Hodgkin’s lymphomas (B-NHL) and compared their levels in CNS-involving lymphomas versus in systemic lymphomas, at diagnosis and during treatment and CNS relapse. We identified a set of five oncogenic microRNAs (miR-19a, miR-20a, miR-21, miR-92a, and miR-155) in CSF that detect, with high sensitivity, secondary CNS lymphoma involvement in aggressive B-NHL, including DLBCL, MCL, and Burkitt lymphoma. Their combination into an oncomiR index enables the separation of CNS lymphomas from systemic lymphomas or nonmalignant controls with high sensitivity and specificity, and high Receiver Operating Characteristics (DLBCL AUC = 0.96, MCL = 0.93, BL = 1.0). Longitudinal analysis showed that oncomiR levels reflect treatment efficacy and clinical outcomes, allowing their monitoring and prediction. In contrast to conventional methods, CSF oncomiRs enable detection of early and residual CNS involvement, as well as parenchymal involvement. These circulating oncomiRs increase 1–4 months before CNS relapse, allowing its early detection and improving the prediction of CNS relapse risk in DLBCL. Similar effects were detectable, to a lesser extent, in plasma.

## 1. Introduction

Central nervous system (CNS) involvement presents a diagnostic and therapeutic challenge in regard to both primary (PCNSL) and secondary CNS lymphomas (SCNSL). Their prognosis is poor and median overall survival (OS) is 4–5 months; 2-year OS is 10–20%, with a less favorable prognosis for SCNSL [1,2].

Secondary CNS lymphomas are characterized by systemic and concomitant or sequential (CNS relapse/progression) CNS-lymphoma involvement. The risk of secondary CNS involvement ranges between 5–25% in all systemic non-Hodgkin lymphomas (NHL) [1,3]. In the most common NHLs, diffuse large B-cell lymphoma (DLBCL, ~40% of NHL) and mantle-cell lymphoma (MCL, ~5% of NHL), the secondary CNS involvement occurs in 5% of cases [4]. In the less common Burkitt lymphoma (BL, 1–2% of NHL), ~25% of systemic BL progresses into the CNS without CNS prophylaxis, and 5% of BL develops into SCNSL with CNS-oriented prophylaxis [5].

Currently, diagnosis of CNS lymphoma involvement is based on conventional methods, such as magnetic resonance imaging (MRI) or computed tomography (CT), followed by brain biopsy, flow cytometry (FCM), cytology, and biochemistry examinations of the cerebrospinal fluid (CSF) [6,7]. Each of these methods suffers from certain limitations—low sensitivity and/or specificity in detecting lymphoma’s spread into the CNS.

Both cytology and FCM are based on the detection of clonal lymphoma cells in CSF, which is limited by the low amount and integrity of these cells in CSF and, in the case of cytology, the difficulty in distinguishing among inflammatory reactive lymphocytes and tumor cells. Notably, these two most commonly used methods are able to detect only meningeal CNS lymphoma involvement. The intraparenchymal lymphomas (30–70%) [4,8] are undetectable by CSF examination and their detection relies on imaging methods (MRI, CT), which can only diagnose already-established tumors (≥0.5–1 cm) with low specificity. Brain biopsy, although specific, is possible only in developed CNS lymphomas detected by imaging methods. The biopsy itself is an invasive procedure with a non-negligible risk of morbidity and mortality (~1%), and may be difficult when deep structures are involved [9]. Altogether, the above-mentioned weaknesses of current detection methods indicate that an early, sensitive, and specific detection method for CNS lymphoma is needed.

Several prognostic models for the risk of CNS involvement have been proposed [4,10,11]. The CNS International Prognostic Index (CNS-IPI) is the most commonly used model for DLBCL, where low-risk patients have less than 1% risk of CNS progression, while in the high-risk group, ~10% of patients develop SCNSL [10]. MIPI is a prognostic index for systemic MCL and can also be applied to predict the risk of CNS involvement [4]. Other risk factors include, for example, lactate dehydrogenase (LDH), β2-microglobulin, and the involvement of multiple extranodal sites (e.g., testes, kidneys). Currently, the dual expression of MYC and BCL2, including their rearrangements (double hit lymphoma), and cell-of-origin (activated B-cell-like) were associated with a higher risk of CNS relapse in DLBCL [8,12,13]. However, the criteria defining high-risk patients receiving CNS-targeted prophylaxis are not uniformly defined. As a result, half of CNS relapses occur outside the high-risk group, and up to 80% of patients in the high-risk group who receive prophylaxis are overtreated. The majority of SCNSL patients subsequently relapse despite reaching complete remission based on routine examination, so the minimal residual disease as the source of subsequent relapses is likely. The sensitive method capable of detecting minimal CNS involvement and/or residual disease would be of benefit for the prediction of CNS relapse/progression and timely application of CNS-oriented treatment or prophylaxis.

MicroRNAs are non-coding RNAs (19–23nt) that negatively regulate gene expression by translational repression and/or mRNA degradation by binding to mRNA 3′UTR [14]. MicroRNAs are often deregulated in tumor diseases, including lymphomas, where they show tumor type-specific expression and can affect tumor biogenesis [15]. MicroRNAs are released into body fluids, where they exhibit extraordinary stability [16,17]. Therefore, circulating microRNAs have become emerging tumor biomarkers. Indeed, several microRNAs, including miR-155, miR-210, miR-21, and miR-17–92, have been shown to be upregulated in sera of systemic lymphomas with prognostic potential [18,19,20]. MiR-21, miR-19b, and miR-92 in CSF [21,22,23,24] and miR-21 in serum [25,26] have been described as having diagnostic potential in PCNSL. However, there are no data on extracellular microRNAs associated with secondary CNS lymphoma involvement.

We evaluated the diagnostic and predictive potential of circulating oncogenic microRNAs (oncomiRs) in CSF and plasma for early and sensitive detection of secondary CNS involvement in aggressive B-NHL, as well as for the detection and prediction of their CNS relapse.

## 2. Materials and Methods

### 2.1. Patients

Paired samples of CSF, plasma, and serum were collected between 2011 and 2019 from 162 B-NHL patients diagnosed with DLBCL, MCL, BL or B-NHL-NOS (not otherwise specified) from two clinical centers: General University Hospital in Prague and University Hospital Kralovske Vinohrady, Czech Republic. The patient cohort included a total of 108 systemic and 54 secondary CNS lymphomas with the following diagnoses: 97 DLBCL (72 systemic, 25 SCNSL), 34 MCL (19 systemic, 15 SCNSL), 18 BL (13 systemic, 5 SCNSL), 13 B-NHL-NOS (4 systemic, 9 SCNSL) and 22 age- and sex-matched control non-lymphoma patients (normal *n* = 8, neurological disorders *n* = 12, FPH *n* = 3, Sjogren syndrome *n* = 2, vascular infarction *n* = 2, encephalitis *n* = 4, multiple sclerosis *n* = 3). The SCNSL cohorts included patients with concomitant CNS involvement at the time of diagnosis of systemic lymphoma (SCNSL-dg) and patients with the sample at the time of detection of CNS relapse (*n* = 20, DLBCL 15, MCL 4, B-NHL-NOS 1), termed “current” relapses. In addition, the cohort included 11 subsequent CNS relapses that occurred during the follow-up of systemic DLBCL (*n* = 7) or DLBCL-SCNSL (*n* = 4) (sampling at diagnosis); in total, the cohort included 32 CNS relapses. Samples were collected at the initial diagnosis and, in selected patients, the serial samples were collected at multiple time-points during the administration of intrathecal therapy or follow-up examinations. Therefore, some patients had a sample from both the diagnosis of systemic lymphoma as well as from CNS relapse. Since lumbar puncture is usually restricted to patients at high risk for CNS disease or patients with corresponding clinical symptoms, the cohort was enriched with patients with a high CNS-IPI score. Patients’ characteristics are shown in Appendix A. This study was approved by the Ethics Committees of both clinical centers. Samples were collected upon receipt of patients’ informed consent.

### 2.2. Sample Processing

Samples were processed within 30–45 min of collection. Until being processed, CSF was kept on ice and plasma and serum samples at RT. CSF was centrifuged at 500× *g*, 10 min, 4 °C to remove cells and debris; plasma and serum samples were centrifuged at 2000× *g*, 15 min at room temperature. Supernatant was snap frozen in liquid nitrogen and stored at −80 °C until further processing. After thawing, samples were homogenized and centrifuged at 500× *g* for 5 min.

### 2.3. MicroRNA Extraction and Quantitation

Total RNA, including small RNAs, was isolated from CSF, plasma, and serum, using an miRNeasy^®^ Mini Kit (Qiagen, Hilden, Germany). Due to the low RNA content in cerebrospinal fluid, the following modifications of the manufacturer’s instructions were used to improve microRNA recovery in all types of samples: (a) 200 µL of sample was mixed with 1 mL of QIAzol^®^ Lysis reagent; (b) Before phase separation, 160 µg of glycogen (Thermo Fisher Scientific, Waltham, MA, USA) and 0.5 pmol cel-miR-39 (spike in control, mirVana^®^, Thermo Fisher Scientific, Waltham, MA, USA) were added to the mix; (c) After adding 200 µL of chloroform, the samples were vortexed vigorously (15–30 s) and centrifuged (15 min, 12,000× *g*, 4 °C); (d) The aqueous phase was mixed with 1.5× volume 100% ethanol, vortexed vigorously, and for enhanced precipitation cooled for 20 min at −20 °C; (e) After-sample temperature was adjusted to RT and the precipitation mix was applied to the miRNeasy^®^ column and centrifuged at 20,000× *g* for 2 min (acceleration 4 to maximize RNA binding); (f) The columns were washed by 1× RWT, 2× RPE buffer and 1× 80% ethanol. Columns were dried by 2 min 20,000× *g* followed by 5 min of RT incubation; (g) RNA was eluted (after 5 min of rehydration) in 40 µL of RNase-free water, supplemented with 1% of RNA inhibitors (Thermo Fisher Scientific, Waltham, MA, USA) using RNA low-binding tubes (LoBind^®^, Eppendorf, Hamburg, Germany).

Reverse Transcription was performed using a TaqMan™ MicroRNA Reverse Transcription Kit and TaqMan MicroRNA Assays (both Applied Biosystems, Waltham, MA, USA), using 10 µL of total RNA in 40 µL of reaction volume (to minimize possible inhibitors concentration) (16 °C 60 min, 42 °C 60 min, 85 °C 5 min). RT-qPCR was performed using a 7900HT Fast real time instrument (Applied Biosystems, Waltham, MA, USA) (40 cycles of 95 °C for 15 s and 60 °C for 1 min). The data were analyzed using SDS 2.0.6 software (Applied Biosystems, Waltham, MA, USA) and relative expression was determined by the 2−ΔCt method using miR-let-7a for normalization (see the Appendix A [18,27]. The efficiency of microRNA isolation and quantification was controlled by an external spike in control (cel-miR-39). Measurements were done in duplicate. Values were equalized to the average value of control samples and presented as a median with interquartile range. The list of used microRNA assays, including a panel for initial screen, is included in the Appendix A.

### 2.4. Statistical Analysis

Group-wise comparisons were performed using Kruskal-Wallis tests with Dunn multiple comparison and Mann Whitney U tests (2-tailed). The specificities, sensitivities, and thresholds (cut-off) were chosen by minimizing the distance of the Receiver Operating Characteristic (ROC) curve to the upper left corner of the unit square: (1 − sensitivity)^2^ + (specificity − 1)^2^ (in most cases equal to the highest Youden index). The Youden index was calculated as the sum of percent of the sensitivity and specificity, minus 100.

The impact of multiple variables, including oncomiR indices, clinical prognostic indices, and other patient characteristics (listed in the Appendix A) on CNS relapse was evaluated by Cox proportional hazards modeling; *p* values were assessed using the log-likelihood ratio test. The event-specific cumulative incidence of CNS relapse and survival probabilities were determined using the Kaplan-Meier method. The end-point of interest was the time to CNS relapse or a death event (OS). For OS, the start-point was Dg of systemic or CNS lymphoma. The groups were compared using the log-rank (Mantel-Cox) test. Details on the stratification of patient groups are in the Appendix A. The statistical and multivariate analyses (for details see the Appendix A) were performed using R statistical software environment version 4.0.2 and GraphPad Prism (Version 5.0, GraphPad, La Jola, CA, USA). The significance level was set to 5% for all analyses. All tests were two-sided.

### 2.5. OncomiR Indces Calculation

OncomiR indices were determined using the logistic regression model (see below) in order to combine abundances of individual microRNAs into a single classifier. For each sample, the oncomiR index was calculated as a sum of individual oncomiRs’ abundances, each multiplied by a coefficient that weighed the prognostic value of a particular oncomiR. The coefficients were obtained from the following general logistic regression formula, using the *solver* tool (MS Excel), by maximizing the difference of average probabilities of CNS involvement between CNS-positive and systemic lymphoma samples:Probability [CNS Involvement] = 1/[1 + exp{X − sum (Coefficient[miRi] × Abundance[miRi])}]

Abundance[miRi] stands for the levels of any of the oncomiRs featuring in the respective index. If the coefficient for one of the oncomiRs was determined to be 0, the respective oncomiR was excluded from the index calculation. MiR-let-7a-normalized and Ctrl/average systemic-equalized data were employed. The indices with oncomiRs combinations with the highest CNS vs. systemic separation (ROC) were finally selected. Further details of coefficient determinations are provided in the Appendix A.

The following oncomiR indices formulae were used (see Appendix A):

CSF:Index [DLBCL] = 1.83 × miR-21 + 1.31 × miR-20a + 1.78 × miR-155
Index [MCL] = 1.36 × miR-21 + 0.83 × miR-20a + 1.30 × miR-92a + 1.84 × miR-155
Index [BL] = 1.57 × miR-21 + 1.75 × miR-155
Index [B-NHL-NOS] = 1.10 × miR-21 + 1.64 × miR-155

Plasma:Index [DLBCL] = 0.07 × miR-21 + 0.03 × miR-19a + 6.93 × miR-20a + 0.43 × miR-155
Index [MCL] = 2.42 × miR-21 + 0.15 × miR-19a + 1.25 × miR-155
Index [BL] = 2.12 × miR-21 + 0.08 × miR-19a + 0.10 × miR-155
Index [B-NHL-NOS] = 5.85 × miR-19a + 0.45 × miR-155

## 3. Results

### 3.1. Oncogenic microRNAs in CSF Enable Detecting CNS Involvement in Aggressive B-NHL

To identify circulating microRNAs that could be used to detect secondary CNS lymphoma involvement, we analyzed lymphoma-associated microRNAs in cerebrospinal fluid (CSF) in aggressive B-NHL subtypes (DLBCL, MCL, BL, and B-NHL-NOS) with and without CNS involvement and compared their levels at diagnosis, during treatment, and during CNS relapse.

Five microRNAs (oncomiRs, miR-21, miR-19a, miR-20a, and miR-92a, miR-155) were selected from 20 candidate microRNAs analyzed in the initial screen (see the Appendix A) according to the highest increase in their levels in the CNS-involving vs. systemic lymphomas (highest CNS lymphoma vs. systemic lymphoma ratio). The candidate microRNAs were selected by a compilation of published data of microRNA expression in CNS involving B-NHL.

The patient cohort included 162 B-NHL patients consisting of 108 systemic and 54 secondary CNS lymphoma (SCNSL): DLBCL = 97 (72 systemic, 25 SCNSL), MCL = 34 (19 systemic, 15 SCNSL), BL = 18 (13 systemic, 5 SCNSL), B-NHL-NOS = 13 (4 systemic, 9 SCNSL). The SCNSL cohorts included patients with CNS involvement at the time of initial diagnosis of systemic lymphoma (SCNSL-dg) and patients with newly detected CNS relapse (*n* = 20, DLBCL = 15, MCL = 4, B-NHL-NOS = 1).

First, we compared CNS-involving lymphomas to a control group with non-lymphoma patients (*n* = 22). All SCNSL subtypes had significantly increased microRNAs of all five varieties (miR-21, miR-19a, miR-20a, miR-92a and miR-155) in CSF (Appendix A, black significance stars). Receiver Operating Characteristic (ROC) analysis revealed high separation and sensitivity/specificity by all five oncomiRs, with the highest by DLBCL-SCNSL miR-21 and miR-20a (AUC = 0.99 and 0.97) and by BL-SCNSL miR-21 and miR-155 (AUC = 1.0 and 0.98) (Appendix A).

For the detection of secondary CNS lymphoma, it is essential to distinguish CNS involvement from systemic-only involvement. Therefore, we compared levels of oncomiRs in CNS involving lymphomas to respective systemic diagnoses. In DLBCL, all tested oncomiRs were significantly increased in SCNSL compared to systemic lymphoma (Appendix A, blue significance stars), which was associated with high CNS vs. systemic separation (highest by miR-21, miR-19a and miR-155: AUC = 0.91; 0.86; 0.85, Appendix A).

Interestingly, the abundance of all tested microRNA in CSF was also increased in CNS-involving MCL, BL, and B-NHL-NOS to a similar (albeit varying) extent as in DLBCL compared to systemic lymphoma (Appendix A), despite lower significances due to a lower number of patients. In MCL-SCNSL, the highest increase and best CNS vs. systemic separation had miR-21, miR-92, and miR-155 (AUC = 0.95, 0.90, and 0.92, respectively, Appendix A). In CNS-involving BL and B-NHL-NOS, there was a significant increase and high CNS/systemic separation of miR-21 and miR-155 (AUC 0.92 and 0.98 for BL and 0.94 and 0.93 for B-NHL-NOS) (Appendix A).

In order to evaluate the abundance of individual oncomiRs as a single variable, we developed a logistic regression model that combines individual microRNA abundances into a single classifier, which we called the oncomiR index (indices). It evaluates the prognostic value of individual microRNAs for particular B-NHL subtypes, to maximize the difference between CNS-positive and systemic lymphomas (see the methods for individual oncomiR contribution).

The oncomiR index overall significantly improved the separation of CNS lymphoma vs. systemic lymphoma in all lymphoma subtypes (AUC: DLBCL = 0.96, MCL= 0.93, BL = 1.0, B-NHL-NOS = 0.94) and provided high sensitivity/specificity (DLBCL 91/90%, MCL 88/93%, BL100/100%, and B-NHL-NOS 100/89%) (see Figure 1A, Figure 2A and Table 1, Appendix A, right panels), indicating that the oncomiR index could be an accurate tool to distinguish CNS lymphoma from systemic lymphoma.

We further subdivided the SCNSL cohorts into patients who already had CNS involvement at the time of diagnosis (SCNSL dg) and patients with newly detected CNS relapse (Figure 1B, oncomiR indices and Appendix A, individual oncomiRs). There was no significant difference between these two SCNSL groups, although a non-significant increase in some oncomiRs in CNS relapses was observed in MCL (Appendix A). 

In addition, we stratified CNS lymphoma with regard to parenchymal and lepto-meningeal involvement. Both lymphoma localizations displayed a comparable significant increase of oncomiR indices (Figure 1C), as well as individual oncomiR levels (Appendix A), compared to the controls, although meningeal and combined meningeal and parenchymal lymphomas had slightly higher median values than parenchymal (1.1–2.1×, all non-significant). This observation indicates that circulating CSF oncomiRs, in contrast to FCM and cytology, are able to detect parenchymal involvement.

In summary, the analysis of microRNA in CSF revealed that five oncogenic microRNAs combined into the oncomiR index can accurately detect secondary CNS lymphoma involvement in all analyzed B-NHL subtypes.

### 3.2. CNS Lymphoma Involvement and CNS Relapse Are Detectable in Plasma

Since collection of CSF is an invasive method, the possibility of testing of oncomiRs in peripheral blood (PB) would be beneficial. Therefore, we investigated the abundance of oncomiRs in plasma (paired samples to CSF). We were able to detect similar trends in the increase of all plasma oncomiRs in all CNS lymphoma subtypes, compared to controls as in CSF (Appendix A, black significance stars). The separation of CNS lymphoma from the control group was in some cases close or equal to 100% (AUC = 1), as reflected by a high sensitivity/specificity (Appendix A).

The oncomiR increase and separation of CNS from systemic lymphoma in plasma was lower than in CSF. Although an increase in oncomiR levels was observed in all cases except for BL, it was less significant (Appendix A, blue significance stars). Similarly, the ability to separate CNS and systemic lymphoma was lower (e.g., DLBCL-SCNSL AUC = 0.76–0.81 vs. 0.85–0.91 in CSF) except for BL-SCNSL, where the separation of miR-21, miR-19a and miR-92a in plasma was better than in CSF (e.g., miR-21 AUC = 1.0) (Appendix A).

**Figure 2 cancers-14-02305-f002:**
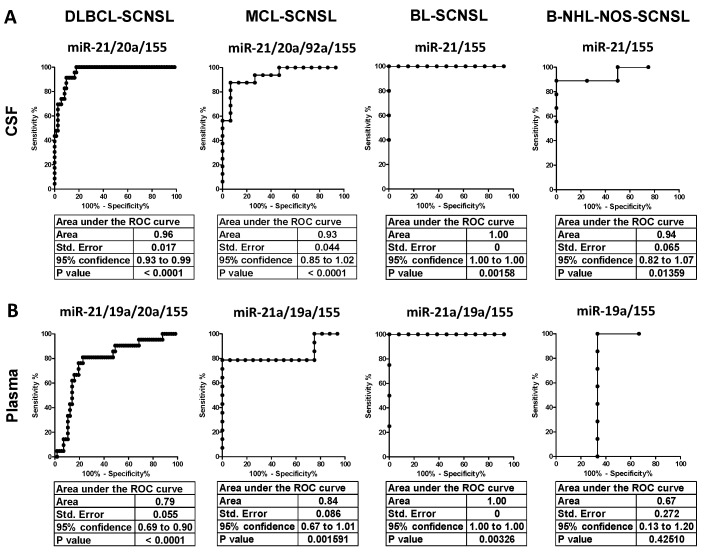
Receiver Operating Characteristics (ROC) of oncomiR indices for discrimination of CNS-involving and systemic lymphomas. (**A**) CSF and (**B**) plasma of indicated CNS lymphoma subtypes. The *X*-axes indicate % of specificity and the *Y*-axes indicate 100—% of sensitivity. The OncomiR index (logistic regression model) combines the expression of individual oncomiRs into a single classifier, yielding higher specificity/sensitivity.

OncomiR indices in plasma reflected individual oncomiRs’ performance. There was a significant increase in oncomiR indices in all CNS lymphoma compared the control group (Figure 3A) and a high separation of CNS lymphoma from control group (DLBCL-SCNSL AUC = 0,94, MCL-, BL- and B-NHL-NOS-SCNSL all AUC = 1, Appendix A), associated with a high sensitivity/specificity (Appendix A, right panels).

However, the increase in oncomiR indices of CNS lymphoma vs. systemic lymphoma was lower in plasma than in CSF (Figure 3). Plasma oncomiR indices had decreased CNS lymphoma vs. systemic lymphoma separation compared to CSF (AUC: DLBCL = 0.79, MCL= 0.84, BL = 1.0, B-NHL-NOS = 0.67), decreased sensitivity/specificity (DLBCL 83/78%, MCL 79/100%, BL100/100% and B-NHL-NOS 100/67%) and *p* value, except for the BL oncomiR index that performed superior CNS vs. systemic separation and sensitivity/specificity (Figure 2 and Figure 3A, Table 1 and Appendix A).

**Figure 3 cancers-14-02305-f003:**
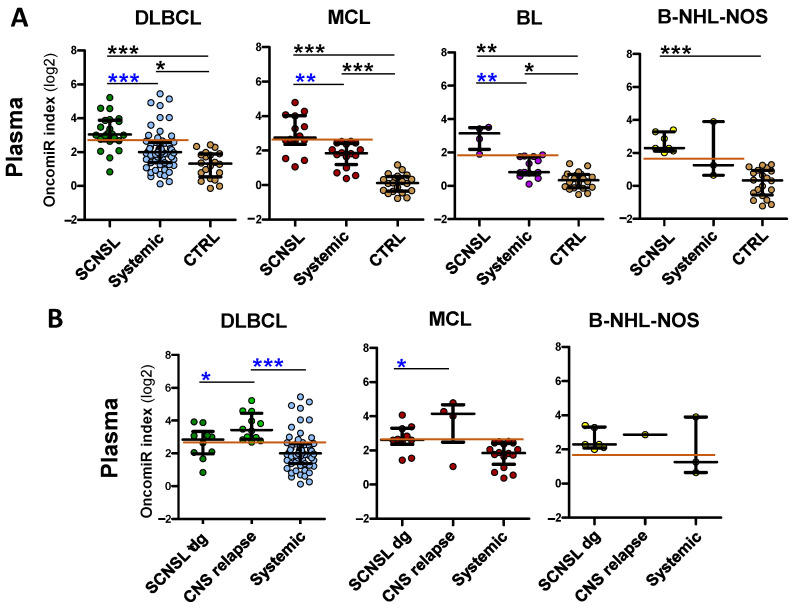
OncomiR indices are increased in plasma of B-NHL involving CNS. OncomiR indices (logistic regression model) combining expression of individual oncomiRs into one classifier in plasma (**A**) of lymphoma patients with indicated B-NHL diagnoses, with (SCNSL) and without secondary CNS involvement (systemic) and control patients (CTRL). (**B**) The SCNSL lymphomas are subdivided into lymphomas with secondary CNS involvement present at diagnosis (SCNSL dg) and newly detected CNS relapses (all BL-SCNSL are from dg). Systemic diagnoses (syst), controls (CTRL). The red line indicates the threshold for positive CNS lymphoma involvement. Log2 scale. Median ± interquartile range, * *p* < 0.05, ** *p* < 0.01, *** *p* < 0.001, Kruskal-Walliss. No star = non-significant.

An interesting phenomenon was observed regarding CNS relapses. There was a significant increase in plasma oncomiR indices (Figure 3B), as well as individual oncomiRs (Appendix A), in DLBCL and MCL CNS relapses compared to patients with CNS involvement already present at diagnosis, which was not observed in CSF. This suggests that the spread of lymphoma from systemic disease to the CNS is likely characterized by an increase (FC 1.5–4×) in plasma oncomiRs.

In summary, we were also able to detect an increase in circulating oncomiR indices in the plasma of CNS-involving lymphomas. While the increase and separation (CNS/systemic) compared to controls were similarly high in plasma as in CSF, they were lower compared to systemic lymphoma, with the exception of BL-SCNSL and CNS relapses (DLBCL, MCL), indicating that plasma can be used for the detection of CNS lymphoma involvement only in these cases.

### 3.3. OncomiRs in CSF and Plasma Reflect Therapy Efficacy and Their Increase Precedes CNS Relapse

We next tested the dynamics of oncomiRs in CSF and plasma during therapy. The CSF and plasma samples were collected at multiple time-points: at the diagnosis, during the administration of intrathecal therapy, or during follow-up examinations.

Interestingly, in **therapy-responders** (*n* = 8), the clinical manifestation of remission was preceded by a gradual decrease in oncomiRs. While FCM and cytology usually switched rapidly from positive to negative findings, the oncomiR levels gradually decreased from diagnosis to remission.

An example of such patients is shown in Figure 4A. In this B-NHL-NOS high-grade secondary CNS lymphoma, the levels of all five oncomiRs in the CSF began to decrease within 3 days after the initiation of therapy, and after a temporal increase in the third last time point, they finally decreased. 

A similar trend was also observed in plasma. Interestingly, an oncomiR decrease in plasma was delayed by several days compared to CSF, and the clearance of plasma microRNAs was preceded by their initial increase (before treatment began to work), indicating different response dynamics in the CNS and the system. Decreasing oncomiR levels in CSF and plasma predicted complete remission ten months in advance.

In contrast, **refractory lymphoma** (*n* = 5) displayed a gradual increase of oncomiRs in CSF during treatment, as documented by the sharp increase of oncomiRs in the Burkitt lymphoma patient (KS IVB, IPI 4) with secondary CNS involvement (multiple systemic + meningeal involvement) (Figure 4B). Despite achieving a clinical partial remission after the first line of treatment, CSF oncomiRs were rising since the beginning of treatment, thus predicting (by 2 months) a later progression that subsequently led to the patient’s death. This trend was visible also in plasma by some (miR-21, 155) but not all microRNA, which is consistent with the fact that of the microRNAs tested, these oncomiRs have a high predictive power in BL plasma (Appendix A). Notably, the initial CNS involvement was not detected by imaging methods (CT), and the later microRNA increase was not reflected by FCM or cytology findings, indicating that oncomiRs better reflected disease development than routine clinical investigation.

The predictive potential of oncomiRs in CSF and plasma is further visible in systemic patients who developed **CNS relapse**. Figure 4C shows a representative example of a systemic DLBCL (*n* = 4) that relapsed/progressed to CNS. This patient (IVB, aaIPI3) achieved a partial remission after first line treatment. Three months later, CNS and systemic progression were confirmed (MRI: parenchymal and meningeal involvement, positive cytology of CSF). The oncomiR levels in plasma initially decreased in response to therapy, while their subsequent increase preceded systemic and CNS progression by 1 month. The high plasma levels of oncomiRs at diagnosis could predict later progression. Despite transient improvement, the patient died 1 month after the last sampling due to the progression of systemic lymphoma. In contrast to plasma, where oncomiRs reflected systemic remission, in CSF the oncomiR levels already increased before and at the time of systemic remission. Thus, the oncomiR increase preceded the lymphoma CNS progression (detected by cytology and MRI) by 3 months. OncomiRs in CSF subsequently decreased with the reduction of CNS involvement. A similar increase in circulating oncomiRs prior to CNS relapse was also observed in the remaining patients.

A total of seven patients with serial samples before CNS relapse (four systemic to CNS relapses and three SCNSL relapses) were detected. In all cases, the gradual oncomiR increase was observed before CNS relapse, in CSF with a median of 3 (range 1–4) months and in plasma 2.7 (1–4) months prior to CNS relapse.

A comparison of the cumulative levels of tested oncomiRs at different stages of the clinical course (Figure 4D) shows that the decrease in oncomiRs is characteristic of partial and complete remission and, conversely, an oncomiRs increase is characteristic of CNS progression and CNS relapse. These trends can be observed in both CSF and plasma.

The longitudinal analysis indicates that: 1. OncomiR levels can change rapidly in response to the disease’s state and therapy; 2. Trends in oncomiR levels reflect and predict therapy efficacy and patient outcomes evaluated by conventional methods; 3. CNS relapse or CNS progression of secondary CNS and systemic lymphoma is preceded by 1–4 months of increasing oncomiR levels in both CSF and plasma.

### 3.4. OncomiRs as a Possible Predictor of CNS Relapse in DLBCL

We hypothesized that in addition to elevated oncomiRs shortly before and during CNS relapse, there is already a predisposition to CNS relapse due to increased levels of CNS-specific oncomiRs at the time of lymphoma diagnosis. To test this hypothesis, we analyzed oncomiR levels in CSF and plasma of systemic DLBCL (*n* = 72) and DLBCL with secondary CNS involvement (SCNSL, *n* = 12) at the time of diagnosis. We compared the subgroup that did not develop CNS relapse with the subgroup that subsequently relapsed to CNS (*n* = 11, systemic to CNS relapse *n* = 7, SCNSL relapse *n* = 4, median time to relapse 9.6, range 4–25 months), which were termed subsequent relapses.

The subsequent CNS relapses of secondary CNS DLBCL (DLBCL-SCNSL) had oncomiR levels among the highest compared to the non-relapsing group, both in CSF and plasma, with a significant increase in plasma miR-19a and miR-92a (Appendix A). In case of subsequent CNS relapse/progression of systemic DLBCL, a significant increase was detected in two microRNAs, miR-155 in CSF and miR-21 in plasma. These data suggest that patients with increased levels of several oncomiRs at the time of diagnosis could be at higher risk of CNS relapse, by both secondary CNS and systemic lymphoma.

Therefore, we assessed the performance of oncomiRs to predict CNS relapse in systemic DLBCL. The multivariate Cox hazards model (with continuous covariates) indicated the significant effect of the oncomiRs on the time to CNS relapse in plasma (*p* = 0.012, miR-21) but not in CSF (*p* = 0.653, miR-155). Therefore, we performed an univariate analysis (Kaplan-Meier) of oncomiRs in plasma. Patients were stratified according to the oncomiR level (high vs. low) at diagnosis (Appendix A). The 4-year cumulative relapse risk in the group with high plasma oncomiR was 31.8% (HR 9.4; *p* = 0.009) versus 6.7% in the low oncomiR group (Figure 5A, Appendix A). For comparison, the CNS-IPI stratified relapse risk was 28.1% for a high CNS-IPI score (Figure 5B). We note that our cohort was enriched with CNS-IPI high-risk patients, since lumbar puncture is limited to patients at high risk of CNS disease or patients with corresponding clinical signs. The multivariate Cox hazards model revealed that oncomiRs were an independent risk factor of CNS-IPI and other clinical characteristics tested (see the Appendix A for multivariate analysis and correlations).

Therefore, plasma oncomiRs and CNS-IPI stratification of relapse risk in DLBCL was combined into one predictive model. This led to identification of a high-risk group with both risk factors (high oncomiR and high CNS-IPI) with a high 4-year relapse rate of 51.5% (HR both vs. no risk = 31.9, *p* = 0.0004, Figure 5C, Appendix A). Notably, the high-risk group accounted for only 17,5% of all patients, compared to 28.6% in the oncomiR model and 42.9% in the CNS-IPI prediction model, thus substantially reducing the high-risk group.

To assess the pre-relapse oncomiR dynamics in DLBCL, we further stratified patients according to the oncomiR index determined at the time of CNS relapse (called current relapse) (DLBCL, N = 15). The relapse rates were higher when compared to the time of diagnosis: CSF 60.6% (HR 31.3) and plasma 55.2%, (HR 16.5) for the high microRNA group (vs. 0% for low microRNA for both CSF and plasma) (Appendix A). This suggests that the predictive power of oncomiRs increases with approaching CNS relapse, consistent with data from the longitudinal analysis.

The prognostic value of oncomiRs was further assessed by the performance of CNS lymphoma-specific oncomiRs to predict overall survival (OS) in DLBCL. The microRNA high/low threshold was prognostic for OS, and patients with high oncomiRs had lower survival (CSF HR = 6.9; *p* = 0.0001, Plasma HR = 2.5; *p* = 0.0014) than patients with low oncomiRs (Figure 6). 

In conclusion, the data indicate that oncomiRs in DLBCL are predictive for CNS relapse (plasma) and OS (CSF + plasma) and the combination of plasma oncomiRs with CNS-IPI significantly improves the prediction of CNS relapse risk.

## 4. Discussion

We focused on finding a sensitive marker of lymphoma CNS involvement in secondary CNS B-NHLs and their CNS relapses. We provide evidence that the expression of five oncogenic circulating microRNAs (miR-19a, miR-20a, miR-21, miR-92a, and miR-155) is elevated to a similar extent in CSF of aggressive B-NHLs with secondary CNS involvement including DLBCL, MCL, and BL. These microRNAs are early and sensitive markers of CNS lymphoma involvement and allow the separation of CNS-involving lymphomas from systemic-only lymphomas. The separation from control samples (nonmalignant neurological disorders) was even higher, which is consistent with earlier reports from PCNSL [21,22,23].

To assess the abundance of tested microRNAs cumulatively, and to account for the variable prognostic power of individual microRNAs in particular lymphoma types, we employed a logistic regression model that combined individual microRNA into a classifier termed the oncomiR index. This approach provided a high separation of the SCNSL vs. systemic lymphoma involvement (AUC: DLBCL = 0.96, MCL = 0.93, BL = 1.0). The high level of CNS/systemic separation allowed the threshold to be set to favor either high sensitivity, specificity, or a balance of the two variables to obtain the optimal diagnostic yield. For example, in the case of DLBCL CSF, the used threshold of 8.2 provided balanced sensitivity (91.3%) and specificity (90,4%), while the threshold 6.9 provided 100%/82% (sensitivity/specificity).

These findings indicate that these circulating CSF oncomiRs, incorporated into the oncomiR index, can serve as an indicator of CNS involvement of aggressive B-NHL (including DLBCL, MCL, and BL), and potentially other types of lymphoma, as supported by similar trends that we observed in follicular lymphoma (not shown). These results are consistent with previous reports showing that two of the five microRNAs tested in this study (miR-21 and miR-92a) are elevated in the CSF of PCNSL [21,22,23].

Although not tested, we suggest that the combination of several oncomiRs may provide specificity to distinguish CNS lymphoma from other diseases involving the CNS. For example, the most common glioblastoma has elevated CSF miR-21 [24] but not miR-155 or any of miR-17-92 cluster. Similarly, miR-21 [25] and miR-20a [26], but not other microRNAs that we tested, were found to be elevated in multiple sclerosis.

Although the studied B-NHLs represent distinct lymphoma subtypes, they are all of B-cell origin and, despite differences, they share common features. All tested oncomiRs have been reported to be upregulated in biopsies or blood of studied systemic lymphoma subtypes. MiR-17-92 was the first described to be upregulated in various lymphoma subtypes [28,29], with miR-19a as a main oncogenic member [30]. MiR-155 was found to be upregulated in DLBCL [31,32,33,34] and BL [35,36] and in the leukemic fraction of MCL [37]. MiR-21 is overexpressed in various malignancies, including B-NHL, of studied subtypes [33,34]. All of these microRNAs are considered to be oncogenes inhibiting numerous tumor suppressors and signaling apoptotic and cell cycle pathways. Moreover, ectopic expression of miR-17-92 and miR-155 can cause lymphoproliferation and B-cell malignancies [38,39], indicating their vital role in lymphoma biogenesis. Interestingly, both miR-17-92 cluster and miR-155 are dose-dependently transcriptionally regulated by hematopoietic transcription factor PU.1 [40,41] that is often deregulated in hematopoietic malignancies [37].

The different oncomiRs included in the oncomiR indices or their varying strengths (represented by coefficients) between different lymphoma subtypes probably reflect molecular differences in the physiology of individual lymphoma subtypes. The differences between CSF and plasma index coefficients likely reflect the differences in the complexities of CSF and plasma, which may reflect the sum of circulating microRNAs from different sources contained in CSF and plasma (plasma also contains microRNAs from other organs and tissues, as discussed below). The determination of oncomiR index coefficients also depends on the number of patients studied, and therefore validation of the study in a larger number of patients may lead to more accurate oncomiR indices (coefficients) and the thresholds for lymphoma CNS involvement.

Using ROC analysis, we determined the threshold values of the oncomiR index for detecting CNS lymphoma involvement. These thresholds are valid for both concurrent systemic and secondary CNS involvement at the time of initial diagnosis of SCNSL, as well as for newly detected CNS relapses. We note that the index thresholds apply only to oncomiR index values, but not to individual microRNAs, and due to the different microRNAs incorporated into the indices or their different coefficients, the thresholds vary between lymphoma subtypes.

Conventional diagnostic methods for CNS lymphoma involvement have limitations. The most commonly used methods, flow cytometry and cytology of CSF, are not able to detect the intra-parenchymal involvement that is detectable only by magnetic resonance (with low specificity/sensitivity) followed by brain biopsy [1,6]. MicroRNA evaluation in CSF overcomes this weakness and detects parenchymal involvement. In addition, oncomiRs could be useful in those CNS lymphomas where biopsy cannot be performed to establish the diagnosis.

Due to the invasiveness of lumbar puncture, the testing of lymphoma CNS involvement by microRNAs from blood would be beneficial. Indeed, we were able to detect increased levels of tested microRNAs in both plasma and serum (data not shown). The relative microRNA abundance in plasma and serum was comparable. We preferred to use plasma samples because there was a higher variability in microRNA levels in the serum (e.g., in subsequent serial samples). We speculate that this is probably due to the fact that extracellular microRNAs are incorporated in protein (e.g., Argonaute) complexes [14,42] which may be differently precipitated and cleared during the blood coagulation of individual samples.

Although the oncomiR increase and separation of CNS lymphoma from control samples was, in plasma, comparable to that in CSF, the separation of CNS from systemic lymphoma in DLBCL, MCL, and B-NHL-NOS was lower in plasma. This finding is possibly caused by several reasons: 1. systemic lymphomas have higher levels of oncomiRs in plasma than in CSF (compared to controls, approximately twice as high); 2. The heterogeneity of lymphoma cells. Our data indicate that the CNS is likely invaded by lymphoma cells/clones with a higher expression of oncomiRs (likely due to their higher aggressiveness); 3. Plasma potentially contains circulating microRNAs produced by other organs (especially those highly perfused). Thus, plasma microRNA levels are complex and reflect the sum of the microRNAs produced by systemic lymphoma, other organs, and potentially the microRNA from CNS lymphoma (if the blood brain barrier is compromised or permissive for microRNA), due to the higher volume of plasma diluted. As a result, we observed a high diagnostic CNS vs. systemic separation in plasma in BL (oncomiR index AUC = 1.0), and acceptable in MCL (AUC = 0.84) but not DLBCL (AUC = 0.78) and B-NHL-NOS (AUC = 0.67).

Longitudal microRNA analysis during the treatment of CNS lymphoma revealed the following phenomena: oncomiR levels in CSF reflect, with high accuracy, the disease state and response to treatment, thus allowing their monitoring. Furthermore, the trends in microRNA expression preceded and thus predicted subsequent remission or progression in advance. In particular, microRNAs gradually decreased in the case of responsive CNS lymphomas and, reciprocally, gradually increased in the case of refractory CNS lymphomas. OncomiRs were able to monitor treatment efficacy at a higher resolution (gradually) and over a longer time window than FCM or cytology; e.g., in advance or after (on average up to 3 months) positive FCM/cytology findings, detecting likely initial or residual disease. In numerous samples, microRNAs were able to detect CNS involvement that was not detected by FCM or cytology of CSF. For example, oncomiRs predicted treatment failure up to 3 months earlier than FCM or cytology (Figure 5B).

Both CSF- and plasma-derived oncomiRs have the ability to monitor the response to treatment (Figure 4A–D). However, we observed that the microRNA trends in CSF and plasma may differ in the case of different development in systemic and CNS diseases. Therefore, we suggest that CSF oncomiRs are more accurate for monitoring CNS diseases.

The sensitivity and timeliness of microRNA evaluation in CSF (and plasma) is further documented by increased oncomiR levels in advance of CNS relapse. First, CNS relapses (DLBCL, MCL, B-NHL-NOS) have significantly increased oncomiRs in CSF and plasma at the time of relapse detection. In addition, long-term analysis showed that oncomiR levels begin to increase 1–4 months prior to the clinical demonstration of the relapse in both CSF and plasma (at times when FCM and cytology were negative), suggesting that increasing oncomiR levels indicate emerging CNS relapse. Thus, circulating oncomiRs enabled the detection of CNS relapse not only at the time of clinical relapse, but also several months in advance, allowing a potentially earlier application of therapy. We speculate that this predictive potential is likely due to the sensitivity of microRNA analysis to detect occult initial stages of CNS involvement, not detectable by conventional methods. It should be noted that while multiple oncomiRs were increased at the time of detection of secondary CNS involvement, this effect decreased with prolonged time before CNS relapse. At the time of diagnosis of systemic DLBCL, several months to several years before CNS relapse, only plasma miR-21 and CSF miR-155 were significantly elevated.

We further explored the ability of oncomiRs to predict CNS relapse at the time of diagnosis of systemic DLBCL. The risk model, based on stratification of patients by high/low oncomiR in plasma, indicated that circulating oncomiRs (as an independent risk factor) could provide a comparable estimate of relapse rate as CNS-IPI (31.8 vs. 28.1%). Interestingly, the oncomiR model identified a lower number of non-relapsing patients in the high-risk group than CNS-IPI (23.2% vs. 37.5%). The predictive oncomiR potential increased with the time to approaching relapse (longitudinal analysis), and oncomiR levels at diagnosis were lower than at the time of CNS relapse. Unlike plasma, the CSF oncomiRs at diagnosis were not significant for CNS relapse prediction. This is consistent with a scenario in which CNS is not initially affected and lymphomas invade the CNS from systemic disease, which is likely predisposed to CNS relapse by higher oncomiRs, and CSF oncomiRs subsequently increase at the time when lymphoma spreads to the CNS.

In addition, we combined CNS-IPI and plasma oncomiR stratification of relapse risk to one prediction model. Notably, the incorporation of oncomiRs into the CNS-IPI/oncomiR model significantly improved the 4-year relapse risk prediction (nearly two-fold), compared to the single CNS-IPI or oncomiR models (51.5% vs. 28.1% and 31.8% respectively). It substantially reduced the high-risk group to only 17% of all patients (compared to 42.9% in CNS-IPI and 28.6% in the oncomiR model), by reducing the non-relapsing patients in the high-risk group. This may help to limit the use of CNS-oriented prophylactic treatment to only those high-risk patients who need it and avoid the over-treatment of patients who cannot benefit from it. A similar approach integrating biomarkers into CNS-IPI was recently described [8]. However, our cohort included limited data on cell-of-origin and MYC/BCL2 expression and rearrangements [12,13] to be employed. In addition, our cohort included a small number of patients with subsequent CNS relapse; therefore, a further validation study is required. It would also be useful to test the potential of CNS-specific oncomiRs to predict CNS relapse in MCL and BL, which currently lack predictive models of CNS relapse risk [4].

Finally, we found that high levels of oncomiRs are prognostic for overall survival (OS) in both CSF and plasma of DLBCL, further confirming the prognostic and CNS lymphoma detection capabilities of circulating oncomiRs.

A summary of the possible use of CSF and plasma oncomiRs as markers of CNS lymphoma involvement is provide in the Appendix A.

## 5. Conclusions

This study describes a set of circulating oncogenic microRNAs in CSF and plasma that (combined into the oncomiR index) can serve as sensitive markers of secondary CNS B-NHL, including DLBCL, MCL, and BL. In contrast to conventional diagnostic methods, these oncomiRs are able to detect early and residual CNS lymphoma involvement, as well as parenchymal involvement in DLBCL. Levels of these oncomiRs reflect and predict clinical responses to therapy. In addition, oncomiRs are increased before CNS relapse and in DLBCL (plasma) oncomiRs improve in combination with CNS-IPI the prediction of CNS relapse risk.

In summary, the study indicates a potential use of microRNA evaluation in CSF and plasma for early detection of secondary CNS involvement in aggressive B-NHL lymphomas, as well as for the monitoring and predicting of therapy efficacy and for the prediction of CNS relapse or its early detection.

## Figures and Tables

**Figure 1 cancers-14-02305-f001:**
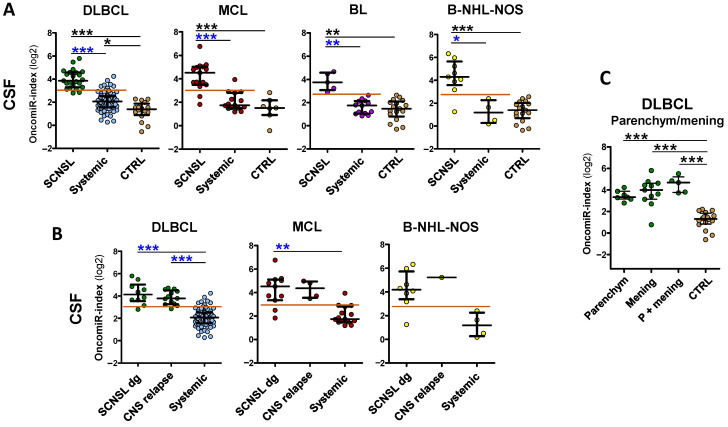
OncomiR indices are increased in cerebrospinal fluid of CNS-involving lymphomas. OncomiR indices (logistic regression model) combining expression of oncogenic microRNAs (oncomiRs, miR-21/-19a/-20a/-92a/-155 as in Table 1) into single classifier in cerebrospinal fluid (CSF) of lymphoma patients with indicated B-NHL diagnoses. (**A**) Lymphoma patients with secondary CNS involvement (SCNSL), compared to systemic lymphoma patients and control patients (CTRL). (**B**) SCNSL lymphomas are subdivided into lymphoma with secondary CNS involvement presented at the time of diagnosis (SCNSL dg) and newly detected CNS relapses. (**C**) OncomiR indices in CSF of DLBCL stratified according to parenchymal, meningeal, and combined parenchymal and meningeal (P+mening) CNS involvement (all BL-SCNSL are from dg). qRT-PCR. The red line indicates the threshold for positive CNS lymphoma involvement. Log2 scale. Median ± interquartile range, * *p* < 0.05, ** *p* < 0.01, *** *p* < 0.001, Kruskal-Walliss. No star = non-significant.

**Figure 4 cancers-14-02305-f004:**
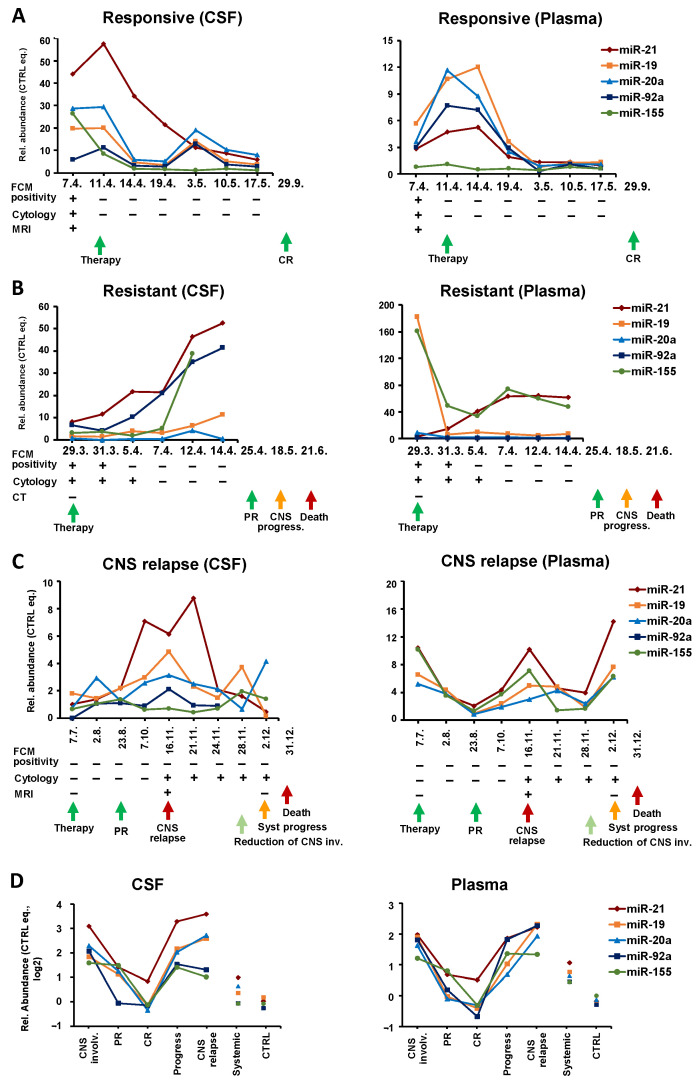
Dynamics of oncomiRs during therapy of CNS lymphoma patients. Levels of indicated oncomiRs in cerebrospinal fluid (CSF, **left** panel) and plasma (**right** panel) of patients during treatment. (**A**) B-NHL-NOS patient with secondary CNS involvement responding to therapy; (**B**) BL patient with secondary CNS involvement resistant to therapy; (**C**) DLBCL systemic patient who relapsed/progressed to CNS. Below the x-axes the following are indicated: date of sampling and positivity/negativity of findings of flow cytometry/cytology of CSF, magnetic resonance imaging (MRI), and computed tomography (CT). (**D**) Median values of oncomiR levels of patients during the course of the disease of CNS-involving lymphomas, compared to systemic lymphomas and non-lymphoma controls. Abbreviations: CNS inv. = CNS involvement at the time of diagnosis; PR = partial remission; CR = complete remission.

**Figure 5 cancers-14-02305-f005:**
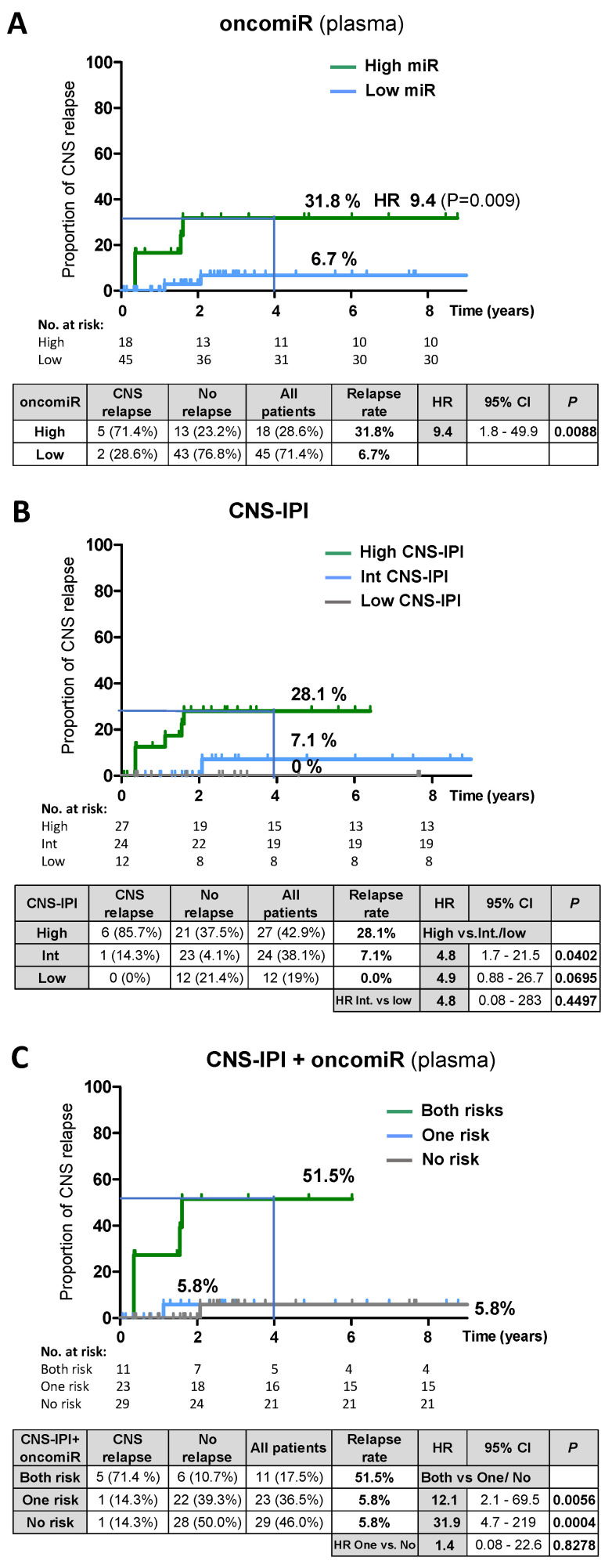
Validation of predictive value of circulating oncomiRs in plasma for CNS relapse in DLBCL. Kaplan-Meier estimates of risk for CNS relapse stratified by: (**A**) oncomiR levels in plasma acquired at the time of diagnosis, (**B**) CNS-IPI, and (**C**) combined prediction model of CNS-IPI and oncomiR risk. Both risks = high microRNA + high CNS-IPI; One risk = either high microRNA or highCNS-IPI; No risk = neither microRNA nor CNS-IPI are high. For details on risk stratification, see the Appendix A. Note: HR, 95% Cl and P in the tables below the charts were obtained from univariate models of indicated categories. Abbreviations: CNS = central nervous system; IPI = International Prognostic Index; Int. = intermediate; HR = hazard ratio; *n* (%) = number of patients.

**Figure 6 cancers-14-02305-f006:**
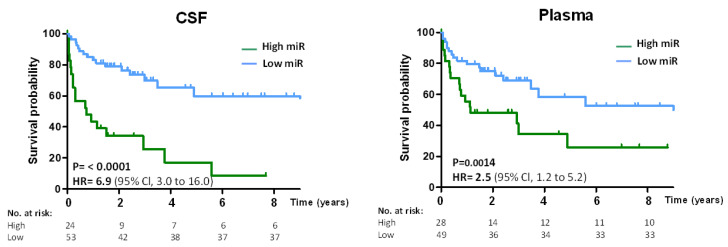
Probability of overall survival in DLBCL according to oncomiR levels in CSF and plasma. Overall survival (OS) Kaplan-Meier estimates of DLBCL patients (with both systemic and concomitant systemic and CNS involvement), stratified by oncomiR index in CSF (**left** panel) or plasma (**right** panel). For details on OS stratification, see the Appendix A.

**Table 1 cancers-14-02305-t001:** Predictive accuracy of oncomiR indices to detect CNS lymphoma involvement.

**CSF**	**oncomiR Index**	** *p* ** **Value**	**AUC**	**Sensitivity**	**Specificity**	**Youden Index**	**Threshold (log2)**
DLBCL-SCNSL	miR-21/20a/155	<0.0001	0.96	91.3	90.4	81.7	8.42 (3.07)
MCL-SCNSL	miR-21/20a/92a/155	<0.0001	0.93	87.5	93.3	80.8	8.86 (3.15)
BL-SCNSL	miR-21/155	0.0016	1.00	100.0	100.0	100.0	6.83 (2.77)
B-NHL-NOS-SCNSL	miR-21/155	0.0136	0.94	88.9	100.0	88.9	7.14 (2.84)
**Plasma**	**oncomiR Index**	** *p* ** **Value**	**AUC**	**Sensitivity**	**Specificity**	**Youden index**	**Threshold (log2)**
DLBCL-SCNSL	miR-21/19a/20a/155	<0.0001	0.79	83.3	78.3	61.6	6.13 (2.62)
MCL-SCNSL	miR-21/19a/155	0.0016	0.84	78.6	100.0	78.6	5.82 (2.54)
BL-SCNSL	miR-21/19a/155	0.0033	1.00	100.0	100.0	100.0	3.68 (1.88)
B-NHL-NOS-SCNSL	miR-19a/155	0.4251	0.67	100.0	66.7	66.7	3.20 (1.68)

*p* value (the effect of oncomiR indices on CNS lymphoma involvement, likelihood ratio test of the multivariate regression analysis), AUC (the area under curve of Receiver Operating Characteristics), specificity, sensitivity, Youden index (the highest sum of specificity and sensitivity—100), and threshold (the cut-off value for lymphoma CNS involvement) all detect CNS lymphoma vs. systemic lymphoma involvement.

## Data Availability

The data presented in this study are available in this article (and Appendix A).

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
