# Peer review of "Circulating microRNAs in Cerebrospinal Fluid and Plasma: Sensitive Tool for Detection of Secondary CNS Involvement, Monitoring of Therapy and Prediction of CNS Relapse in Aggressive B-NHL Lymphomas"

_cancers, 2022, doi:10.3390/cancers14092305_

Round 1
Reviewer 1 Report
Authors have addressed the comments very nicely. Paper can be accepted now
Reviewer 2 Report
Authors argued and have modified the mansucript according to the comments adressed. Few comments:
- OS charts of MCL should be added in the supplement figures.
- line 245: n=55 secondary CNS. This is different from methods section (n=54)
- line 247: n=15 SCNSL instead of n=13 in method section. and B-NHL= 13 and not 12 (9+4)
- -line 340-341, reformulate...
- - line 493: tipo: "prediction"
- - line 544: tipo " included"
- - line 587: "BBB"?
Reviewer 3 Report
These comments however do not address the issues directly and neither do they alter my primary conclusion - that this method is not going to be useful clinically. Arbitrary selection of miRNAs is not acceptable Tumours have not been characterised adequately Therefore my opinion on this MS is unaltered - personally I don’t think that these data merit publication
Reviewer 4 Report
Authors have properly addressed my comments.